Anticancer drug synergy prediction based on CatBoost

Li Changheng
Guan Nana gnn@mail.gufe.edu.cn
Zhang Hongyi
College of Big Data Statistics, Guizhou University of Finance and Economics , Guiyang , China
Chicco Davide
Electronic publication date: 2025 May 19
Publication date: 2025
Volume: 11
Electronic Location ID: e2829
Received 2024 Jun 12; Accepted 2025 Mar 24
Copyright: © 2025 Li et al.
Copyright year: 2025
Copyright holder: Li et al.
License: This is an open access article distributed under the terms of the Creative Commons Attribution License, which permits unrestricted use, distribution, reproduction and adaptation in any medium and for any purpose provided that it is properly attributed. For attribution, the original author(s), title, publication source (PeerJ Computer Science) and either DOI or URL of the article must be cited.
License URL: https://creativecommons.org/licenses/by/4.0/

Keywords: Anticancer drug, Drug synergy, Prediction, CatBoost

Funding: Science and Technology Planning Project of Guizhou Province of China Qian Ke He Ji Chu -ZK[2021] Yi Ban 315 Science Foundation of Guizhou University of Finance and Economics 2021KYYB21 Youth Science and Technology Talent Growth Project of Guizhou Provincial Education Department No. QJH-KY-Z[2021]132 This work was supported by the Science and Technology Planning Project of Guizhou Province of China (No. Qian Ke He Ji Chu -ZK[2021] Yi Ban 315), Science Foundation of Guizhou University of Finance and Economics (No. 2021KYYB21) and the Youth Science and Technology Talent Growth Project of Guizhou Provincial Education Department (No. QJH-KY-Z[2021]132). The funders had no role in study design, data collection and analysis, decision to publish, or preparation of the manuscript.

==============================
Background

The research of cancer treatments has always been a hot topic in the medical field. Multi-targeted combination drugs have been considered as an ideal option for cancer treatment. Since it is not feasible to use clinical experience or high-throughput screening to identify the complete combinatorial space, methods such as machine learning models offer the possibility to explore the combinatorial space effectively.

Methods

In this work, we proposed a machine learning method based on CatBoost to predict the synergy scores of anticancer drug combinations on cancer cell lines, which utilized oblivious trees and ordered boosting technique to avoid overfitting and bias. The model was trained and tested using the data screened from NCI-ALMANAC dataset. The drugs were characterized with morgan fingerprints, drug target information, monotherapy information, and the cell lines were described with gene expression profiles.

Results

In the stratified 5-fold cross-validation, our method obtained excellent results, where, the receiver operating characteristic area under the curve (ROC AUC) is 0.9217, precision-recall area under the curve (PR AUC) is 0.4651, mean squared error (MSE) is 0.1365, and Pearson correlation coefficient is 0.5335. The performance is significantly better than three other advanced models. Additionally, when using SHapley Additive exPlanations (SHAP) to interpret the biological significance of the prediction results, we found that drug features played more prominent roles than cell line features, and genes associated with cancer development, such as PTK2, CCND1, and GNA11, played an important part in drug synergy prediction. Combining the experimental results, the model proposed in this study has a good prediction effect and can be used as an alternative method for predicting anticancer drug combinations.

Introduction

Cancer is a huge threat to the health of all mankind. Chemotherapy has been a common strategy for cancer treatment for a long time, but it has proven to be associated with many side effects (Madani Tonekaboni et al., 2018). Reports have shown that the cancer monotherapy often suffers from limited efficacy, poor safety, and drug resistance (Andreuccetti et al., 1996; Lehár et al., 2009). Drug resistance and side effects have been the main reasons for the failure of cancer chemotherapy. On the other side, the progress of cancer drug research and development has become slower, and the cost of developing new drugs has become higher (He et al., 2018). Therefore, it is a big challenge to improve the efficiency and reduce the cost of drug research and development.

With the arising of network pharmacology, multi-target combination drugs have become a new research direction (Gottesman, 2002; Holohan et al., 2013; Siegel, Miller & Jemal, 2019). Combination drugs may have greater or lesser effect on cancer cells than the additive sum of their individual effects, i.e., synergistic or antagonistic effects (Madani Tonekaboni et al., 2018). The synergistic drug combinations usually need lower dose than single drugs, with improved efficacy and reduced drug toxicity. Besides, they can delay the development of drug resistance to the greatest extent. Therefore, drug combinations with synergistic effects may be the ideal therapeutic regimens for cancer (Chou, 2006; Csermely et al., 2013). Finding synergistic drug combinations for specific cancer types is important to improve the efficacy of anticancer therapy (Huang et al., 2017; O’Neil et al., 2016).

Effective drug combinations can be proposed based on clinical experience, but the benefits of this approach are much less than the time and cost it consumes. Another strategy to identify synergistic drug combinations is high-throughput screening (HTS) (Jia et al., 2009). The HTS method can yield a large number of experimental results in a reasonable time and at a low cost, making it the preferred choice for discovering effective combinations of drugs. Finding new effective drug combinations is a complex task because the number of possible drug combinations is very large and this number increases each time a new drug is developed. It is clearly not feasible to identify the complete combination space using HTS.

As the data from clinical experience and high-throughput screening accumulating, opportunities for large-scale application of machine learning methods are available. For example, AstraZeneca, a leading pharmaceutical company, partnered with several organizations to launch a drug combination prediction challenge in the DREAM community, providing participants with 11,576 synergistic data derived from 910 drug combination experiments, involving 118 drugs and 85 cancer cell lines (Menden et al., 2019). In 2017, the National Cancer Institute (NCI) released the largest of publicly available cancer drug combination datasets, ALMANAC, which contains synergy measurements for drug combinations of 104 drugs in 60 cancer cell lines in NCL-609. A web-based portal named DrugComb (https://drugcomb.org/) have been built and widely used in the research of drug combination (Zagidullin et al., 2019; Zheng et al., 2021). Based on these data resources, several machine learning algorithms for anticancer drug combination prediction have been proposed (Wu et al., 2021). For instance, Li et al. (2017) used the data from DREAM to predict drug combination synergy using a random forest model based on drug-target networks and gene expression profiles. Besides, Li et al. (2018) proposed a novel network propagation method to simulate molecular features based on gene-gene networks and drug-target information, and combined the molecular features with single-drug treatment data to train random forest as a classifier for anticancer drug synergy prediction. Janizek, Celik & Lee (2018) proposed a method based on Extreme Gradient Boosting (XGBoost) to predict drug combination synergy. Celebi et al. (2019) also proposed an XGBoost based approach to predict anticancer drug combinations using multi-omics data. In their work, the targeting pathways and monotherapy information were added to the feature space. Sidorov et al. (2019) used XGBoost as well as random forest to build a separate model for each cell line, for the prediction of synergistic effects of anticancer drug combinations. Jeon et al. (2018) proposed an extremely randomized trees (ERT)-based method for predicting anticancer drug combinations. Li et al. (2020) used logistic regression to test the statistical significance of gene and pathway features in predicting the synergy of anticancer drug combinations. Julkunen et al. (2020) proposed a new prediction method called ComboFM, which models the auxiliary features of two drugs, cell lines, and drug-cell lines as a fifth-order tensor and predicted the response of drug pairs using higher-order factorization machines (HOFM). With the development of deep learning algorithm, more and more models were constructed for drug synergy prediction based on deep learning. Preuer et al. (2018) proposed a model named DeepSynergy, which is a three-layer feedforward neural network using genomic information and drug-chemical features as input features. They used a normalization strategy to account for the heterogeneity of the input features, and a conical layer model to predict drug synergy. Besides, Zhang et al. (2020) proposed the model DeepSignalingSynergy. Instead of considering a large number of chemical and genomic features, the authors only utilized a small number of cancer signaling pathways to investigate the importance of individual signaling pathways for prediction. Zhang et al. (2021) proposed the model AuDNNsynergy to predict drug combination synergy by integrating multi-omics and chemical structure data. Kim et al. (2021) developed a drug synergy prediction model based on multitasking deep neural networks integrating multimodal inputs and multimodal outputs using data from multiple cell line features, and used migration learning to study data-poor tissues using data-rich tissues. Wang et al. (2022) proposed a new deep learning prediction model PRODeepSyn. The model used graph convolutional neural networks to integrate protein-protein interaction networks (PPI) and histology data to construct low-dimensional embeddings of cell lines, which were fed into the deep neural networks together with the drug features to calculate drug synergy scores. Similarly, Hu et al. (2022) proposed DTSyn to understand the mechanism of drug synergy from the perspective of chemical-gene-tissue interactions. Gan et al. (2023) proposed a deep learning model, DGSSynADR, to predict synergistic anticancer drug combinations, which performed a low-rank global attention model on a heterogeneous graph for feature construction. Recently, Huangfu et al. (2024) proposed SNSynergy, a machine learning framework based on a locally weighted model that predicts synergy scores using a small amount of auxiliary data by analyzing similarities between cell lines and drug combinations. The drug combinations identified by SNSynergy fit with existing studies and were partially validated in the clinic, demonstrating its potential in high-throughput screening. Ren et al. (2024) developed an XGBoost feature fusion model to predict effective drug combinations for a specific cancer by combining drug molecular features and cancer-related gene information. The model makes use of disease-specific data and improves the accuracy of prediction. Besides, Liu et al. (2024) proposed a graph neural network model named SynerGNet for drug synergy prediction based on the cancer-specific featured graphs.

In this work, we present a CatBoost-based machine learning approach to predict the synergy scores of anticancer drug combinations. CatBoost is a symmetric decision tree (oblivious trees) based learner implementation with fewer parameters, which supports category-based variables and high accuracy gradient boosted decision tree (GBDT) framework. CatBoost has been widely used in the biomedical field for various tasks and studies. In a recent study, Pudova et al. (2023) utilized the CatBoost algorithm to identify cancer-related microRNAs (miRNAs). Jinchao et al. (2020) proposed a prediction model called CatBoost-SubMito for protein submitochondrial location prediction. In addition, Bouget et al. (2022) used the CatBoost algorithm to predict patients’ responses to tumor necrosis factor inhibitors. Clearly, CatBoost is playing an important role in the biomedical field.

In our previous work, based on the CatBoost algorithm, we explored ways to use potential features of drugs to predict drug combinations (Li, 2024). In this article, we used CatBoost to construct a prediction model based on features of drugs and cell lines. The performance of CatBoost was evaluated using stratified 5-fold cross-validation. We found that CatBoost outperformed the models based on deep neural networks (DNN), XGBoost and logistic regression in all metrics. In addition, an interpretation package named SHapley Additive exPlanations (SHAP) was introduced to interpret the biological significance of the prediction results. It was found that the top-ranked genes contributing to the CatBoost model predictions were almost associated with known cancer mechanisms. Portions of this text were previously published as part of a preprint (https://doi.org/10.21203/rs.3.rs-3652163/v1) (Li, Guan & Zhang, 2023).

So far, few scholars have applied CatBoost algorithm to the field of anticancer drug combination prediction. CatBoost has the advantages of few parameters, excellent performance and robustness, which can be applied to the field of anticancer drug combination synergy prediction. In this work, multiple features of cell lines and drugs were fed into the CatBoost to improve its prediction performance. Besides, a model interpretation package called SHAP was used to analyze the biological significance behind the prediction results.

Materials and Methods

Datasets

We used the datasets from reference (Fan, Cheng & Li, 2021), which considered the NCI-ALMANAC (https://wiki.nci.nih.gov/display/NCIDTPdata/NCI-ALMANAC) dataset as the source of anticancer drug synergy data. NCI-ALMANAC contains data of 59 cancer cell lines and we only considered drugs that have at least one target gene (68 drugs). A total of 130,182 samples were used for model training and testing. All of the NCI-60 cancer cell line characteristics (expression, mutations, copy number variant (CNV), etc.) could be obtained from CellMinerCDB (https://discover.nci.nih.gov/rsconnect/cellminercdb/) (Luna et al., 2021). Drug target data was provided by DrugBank (Wishart et al., 2018), and the drug molecular properties could be processed using the RDKit package in Python. Cell line characteristics generally include gene expression profiles, mutation numbers, copy numbers, etc., while drug characteristics usually include morgan fingerprints, drug target information, monotherapy information, etc. In this work, we used gene expression profiles as cell line features, and morgan fingerprints, drug target information, and monotherapy information as drug features. The shape of the dataset is 130,182 samples with 2,064 columns of features, of which 470 are cell line features and 1,594 are drug features.

We used Python’s pandas package to process the data. Firstly, pandas is used to extract four columns of data (drug1, drug2, cell line, score) required in this article, which are drug1, drug2, cell line and synergy score of anticancer drugs. This is a matrix with 130,182 rows and four columns, which is named as the master matrix in this article. Next, through the constructed master matrix, the cell line features are constructed in this article. The gene expression profile contains the gene expression profiles of 470 genes under 59 cell lines, from which a 59 × 470 matrix was constructed in this article.

Then, a zero matrix with 130,182 rows and 470 columns was constructed. Then, the cell lines in the first row of the main matrix were used to get the corresponding row of the gene expression profile matrix, and then assigned to the first row of the zero matrix. The construction of drug features is roughly the same as above, and since each group of samples contains two drugs, the drug features of the two groups of drugs need to be merged. After obtaining all the cell line features and drug features, the NumPy concatenate function is used to concatenate the data by column (axis = 1) to obtain the complete feature data matrix.

CatBoost

Based on the constructed features, this work used a new GBDT framework named CatBoost to predict drug synergy (Fig. 1). CatBoost, as an oblivious trees-based learner, has fewer parameters, and supports for categorical variables, as well as high accuracy GBDT framework. Compared with other methods, CatBoost does not require manual one-hot encoding when processing categorical data, which greatly simplifies the data preprocessing process. In addition, CatBoost reduces the risk of overfitting by using a technique called “ordered boosting” and improves the generalization ability of the model by using a symmetric tree structure. Existing methods, such as XGBoost and LightGBM, although also very powerful and popular, may require additional preprocessing steps when dealing with categorical features, such as manual feature encoding. Moreover, these algorithms may require more hyperparameter tuning to achieve the best performance. As for why CatBoost was chosen over LightGBM, although LightGBM may outperform CatBoost in training speed, CatBoost performs better in terms of generalization accuracy and area under the curve (AUC) on a specific dataset, although this difference may be small. CatBoost’s native support for categorical features and its ability to reduce model overfitting make it potentially a better choice in some cases. In addition, the design of the algorithm structure of CatBoost helps to reduce the gradient boosting bias and prediction offset, thus improving the accuracy of the model. Algorithm 1 gives the pseudo-code of the GBDT algorithm.

Figure 1 The flowchart of the method for anticancer drug synergy prediction based on CatBoost, using drug and cell line features.

Algorithm 1 The gradient tree boosting algorithm.

1. Initialize: f0(x)=arg minγ∑i=1NL(yi,γ).	
2. For m=1 to M :	
3. For i=1,2,…,N compute	
rim=−[∂L(yi,f(xi))∂f(xi)]f=fm−1	
4. Fit a regression tree to the targets rim giving terminal regions Rjm,j=1,2,…,Jm.	
5. For j=1,2,…,Jm compute	
γjm=argminγ∑xi∈RjmL(yi,fm−1(xi)+γ)	
6. Update: fm(x)=fm−1(x)+∑j=1JmγjmI(x∈Rjm).	
7. Output: f^(x)=fM(x).	

Compared with the traditional decision trees, the oblivious trees can better deal with the continuous attribute and the classification problem in the high dimensional space, which improves the prediction ability of the model. Since the oblivious trees can completely ignore some attribute information, it is very small for the existence of noise and missing values in the data. In addition, the oblivious trees use the principle of symmetry, and through the restriction of depth and number of branches, it can effectively avoid the problem of overfitting, and improve the model’s generalization ability. CatBoost also uses the concept of ordered boosting, a permutation-driven approach, which trains the model on a subset of the data while computing the residuals on another subset, thus it can prevent target leakage and over fitting. Algorithm 2 gives the pseudo-code of the ordered boosting algorithm.

Algorithm 2 Ordered boosting algorithm.

1. input: {(xk,yk)}k=1n,I;	
2.  σ←randompermutationof[1,n];	
3. for t←1toIdo	
       for i←1tondo	
          ri←yi−Mσ(i)−1(xi);	
      for i←1tondo	
         ΔM←	
          LearnModel((xj,rj):σ(j)≤i);	
          Mi←Mi+ΔM;	
4. return Mn	

In a nutshell, CatBoost has the following peculiarities: 1. excellent performance, it surpasses most advanced machine learning algorithms in terms of performance; 2. robustness, it reduces the need for much hyperparameter tuning and decreases the chance of overfitting, which makes the model more versatile; 3. practicality, it can handle both categorical and numerical features; 4. scalability, it supports customized loss functions.

The code and datasets of this work could be obtained from: https://github.com/AnnaGuan/CatBoost (DOI: 10.5281/zenodo.14189227).

Experimental setup

To make CatBoost generalizable to unseen datasets, we used stratified k-fold cross-validation experiments to test the model. Stratified k-fold cross-validation is an enhanced version of k-fold cross-validation for unbalanced datasets. In stratified k-fold cross-validation, the whole dataset is divided into k equal-sized copies, and the positive and negative ratio of the label variable in each fold is the same as the percentage in the whole dataset. This work used stratified 5-fold cross-validation to evaluate the model performance. The dataset of all samples was divided into five equal and unique parts, of which four parts were treated as training data and the remaining one as test data. Each part was regarded as test data in turn, and its synergy scores were calculated using the learned model.

By using the stratified 5-fold cross-validation, we also adjusted the parameters of CatBoost to obtain the most optimal model. The main parameters we adjusted included the iterations in the range of [400, 500, 600, 700, 800, 900, 1,000], the depth in the range of [5, 6, 7, 8, 9, 10, 11], and the learning rate in [0.05, 0.5, 0.1]. By comparing the model prediction results under each parameter combination, the optimal parameter combination to train the model was set as: iterations = 600, depth = 9, learning rate = 0.1.

Drug synergy prediction usually has two ways: one is regression task and another is classification task. In this article, we have considered both of these two pattens. When conducting synergy prediction as a classification task, the synergy scores were binarized, i.e., synergy labeled as 1 and antagonism as 0. Since there are many samples with synergy scores close to 0, it is crucial to choose an appropriate threshold to binarize the synergy score. In this work, we optimized the threshold by implementing stratified 5-fold cross-validation to achieve optimal balance, and the value of threshold was finally set as 10.

To evaluate the prediction performance of CatBoost, we compare its synergy prediction ability with the following models: 1. deep neural network (DNN), a deep learning method based on feedforward neural networks; 2. XGBoost, a classical integrated learning method based on gradient boosting; 3. logistic regression, a log-linear regression model based on the sigmoid function. The hyperparameters used in these models were determined by stratified 5-fold cross-validation.

The following statistical measures and methods are mainly used in this article: 1. receiver operating characteristic area under the curve (ROC AUC): it is mainly used for classification tasks to measure the performance of a classifier. 2. Precision area under the recall curve (PR AUC): this is used to evaluate the performance of a classification model, especially when the dataset is imbalanced. 3. mean squared error (MSE): a measure of the average of the squared differences between the predicted and actual values, used for regression tasks. 4. Pearson correlation coefficient (PCC): this is used to assess the linear relationship between the predicted and actual values.

All the experiments were done on Windows 11 on a work computer with the CPU model: 11th Gen Intel (R) Core (TM) i5-11300H @ 3.10 GHz and GPU: Intel (R) lris (R) Xe Graphics.

Results

Performance evaluation

We measured the prediction performance of our model in terms of the receiver operating characteristic area under the curve (ROC AUC) and the area under the precision-recall area under the curve (PR AUC) for classification. To further characterize the prediction performance of CatBoost, we also provided typical performance metrics for regression tasks: mean square error (MSE) and Pearson correlation coefficient (PCC). After carrying out stratified 5-fold cross-validation based on the optimal parameter setting mentioned above, we can compute the values of four metrics for our model and other methods for comparison. The results were shown in Table 1, from which we found that CatBoost outperformed the other three models on all metrics. For example, by comparing the ROC AUC values, CatBoost achieves 0.9217, improved 0.0099 over the next best model DNN; by comparing MSE, CatBoost obtained 0.1365 that was improved 6% over the next best model XGBoost.

Table 1 Comparison of the performance results of four models.

Models	ROC AUC	PR AUC	MSE	Pearson	
CatBoost	0.9217	0.4651	0.1365	0.5335	
DNN	0.9118	0.3876	0.1441	0.4761	
XGBOOST	0.8856	0.3601	0.1439	0.4552	
Logistic regression	0.8505	0.1945	0.1534	0.3101	

To further investigate the quality of CatBoost predictions, we compared the distribution of synergy scores on cell lines between the predicted result and the actual values (seen in Fig. 2). Panel (a) shows the box plots of the distribution of the actual synergy scores for each cell line, and panel (b) shows box plots for the synergy scores predicted by CatBoost. Each point represents the actual or predicted synergy score for a drug combination. To understand how well our model predicted synergy for drug combinations on individual cell lines, we also gave the average PCC between predicted and actual synergy scores for each cell line (Fig. 2C). The order of the cell lines in Fig. 2C is the same as in panel (a) and (b). From panel (c), we could observe that the correlation between all cell lines ranged from 0.52 to 0.83.

Figure 2 The comparison of distribution of synergy scores on individual cell lines between actual data and predicted result.

(A) is the box plots of the distribution of the actual synergy scores for each cell line; (B) is the box plots of the distribution of synergy scores predicted by CatBoost for each cell line; (C) the Pearson correlation coefficient between actual and predicted synergy scores for each cell line.

Additionally, by calculating the PCC, we analyzed the performance of CatBoost from two viewpoints, anticancer drug and cancer cell line, respectively. Figure 3 gives the PCC distribution over 68 drugs, where each bar represents the average PCC of all combinations of the specific drug tagged below the bar. The color of the bars corresponds to the drug target. As shown in Fig. 3, the PCC values for all anticancer drugs ranged from 0.51 to 0.88, where 31% of the drugs have PCC values below 0.6% and 39% above 0.7. We also found that the color of the bars is scattered, which means there is no clear association between the PCC and the drug target. Therefore, the differences of performance between drugs could not be explained by target-based mechanisms. Moreover, the number of drugs acting on the same target does not affect the performance.

Figure 3 The Pearson correlation coefficient distribution over drugs.

Each bar is the average PCC between actual and predicted synergy scores for all combinations of the specific drug whose name is shown on the x-axis. The color of the bars corresponds to the drug target.

The PCC distribution over cell lines is illustrated in Fig. 4. Each bar is the average PCC between actual and predicted synergy scores for a cell line. The cell line names are shown on the x-axis. The color of the bars corresponds to the type of tissue from which the cell line originated. From this figure we can see that PCC values for all cell lines vary from 0.52 to 0.83, with only three cell lines having values below 0.6. Among them, more than 44% of the cell lines have PCC above 0.7. Furthermore, the colors of the bars are scattered and no correlation between tissue type and PCC can be observed. Therefore, the performance differences could not be explained by the tissue type of cell lines either.

Figure 4 The Pearson correlation coefficient distribution over cell lines.

Each bar is the average PCC between actual and predicted synergy scores for a cell line. The names of cell lines are shown on the x-axis. The color of the bars corresponds to the type of tissue from which the cell line originated.

Based on the above analysis, this article argues that the difference in prediction performance is not clearly related to the tissue-specific mechanism and the tissue type/target coverage in the dataset. These differences may be due to the fact that some biological mechanisms are better modeled than others. This may be related to the measurements obtained from cell lines in this article and whether they are able to capture these biological processes.

Feature interpretability

While it is important for a model to predict drug combinations with synergistic effects, it is also important to explain why the model could predict synergistic drugs. Explainable machine learning has slowly become an important research direction in machine learning in the past few years. SHAP is a model interpretation package developed in Python, which can interpret the output of machine learning models. Inspired by cooperative game theory, SHAP constructs an additive explanatory model in which all features are considered “contributors”. For each prediction sample, the model generates a predicted value, and the SHAP value is the number assigned to each feature in the sample, which can be calculated as follows:

(1) SHAPvalues=∑s⊆M{i}|S|!(|M|−|S|−1)!|M|!×f(S∪{i}−f(S))

where M is the set of features, S is a subset of M excluding feature i, and f(S) is the prediction of the model on the feature set S.

For binary classification, we can verify the SHAP calculation using the following formula:

(2) basevalues+∑SHAPvalues[0]=ln⁡(p1−p).

In this work, the SHAP values of all sample features were calculated by SHAP, and the 100 features with highest scores were obtained (see Table S1). We found that among the top 100 features, monotherapy information ranked first, suggesting that monotherapy information was more useful for predicting anticancer drug combination synergy. Statistically, 88 out of these 100 features belong to drug, of which Morgan’s molecular fingerprint takes the largest proportion. The remaining 12 important features were genes in gene expression profiles, including PTK2, CCND1, GNA11, CRKL, ERBB2, WNT2B, CTBP2 etc. In order to investigate the importance of cell line features and drug features, we implemented a comparative experiment through using only cell line features or only drug features respectively for model training and predicting, the results of which are shown in Table 2. We can see that the results of the model using only cell line features are much worse than those using only drug features, both of which are not as good as the original results. Apparently, drug features play a more prominent role in drug synergy prediction than cell line features. The conclusion we reached is consistent with the findings of Janizek, Celik & Lee (2018).

Table 2 Performance comparison of models using only cell line features or only drug features.

Models	ROC AUC	PR AUC	MSE	Pearson	
CatBoost	0.9217	0.4651	0.1365	0.5335	
CatBoost_cell line	0.6686	0.0609	0.1601	0.1262	
CatBoost_drug	0.9148	0.4515	0.1370	0.5287	

For the genes ranking by the front, we have made a case study to further investigate the significance of our model. For instance, PTK2, also known as FAK, was found to associate with tumors and functions primarily in the inhibition of apoptosis (Yi et al., 2021). The upregulation of FAK expression, including high protein expression as well as overactivation, is present in almost all tumor tissues, such as lung, gastric, colorectal, uterine, and melanoma cancers. According to studies (Chuang et al., 2022; Xie et al., 2022), patients with high FAK expression have a significantly shorter survival than those with lower FAK expression, making FAK a promising indicator for early diagnosis of cancer and for predicting the prognosis of patients.

The gene CCND1, or cell cycle protein, plays a role in regulating CDK kinase (Akervall et al., 2003). This protein interacts with the tumor suppressor protein Rb that could positively regulate the expression of this gene. Mutations, amplification and overexpression of this gene alter the course of the cell cycle, which are frequently observed in a variety of tumors, and may contribute to tumorigenesis (Hosokawa & Arnold, 1998).

ERBB2, or erb-b2, is a 185-KDA cell membrane receptor encoded by the proto-oncogene erbB-2 (Penuel et al., 2001). Clinically, erb-B2 expression is strongly correlated with patient prognosis, and patients with high erb-B2 expression are prone to tumor metastasis and have short survival. Because of the significant difference in expression levels between normal and tumor cells, erb-B2 has become an ideal target for tumor immunobiotherapy and is currently a hot molecule in the field of tumor therapy research (Mcdonagh et al., 2012).

To sum up, the top-ranked genes all play important roles in cancer, and such roles are also extended to the field of anticancer drug combination prediction and captured by the model in this article, making an important contribution in the prediction. Obviously, the prediction results of the model in this article have certain biological significance.

Feature reduction

In order to further verify the impact of the front-rank features with high SHAP values on the model efficacy, we selected the top 400 features as input features and performed a new prediction using CatBoost, which was written as new CatBoost. The new CatBoost experiment has the same settings as the previous CatBoost experiment, with the only difference being the number of features used. Considering the balance between computing speed and performance, we chose the top 400 features instead of less, since too few features would result in performance degradation. This feature selection method can be applied to other data sets. The verification experiments were conducted both in stratified 5-fold cross validation and stratified 10-fold cross validation. The prediction performance of new CatBoost is given in Table 3. By comparison, it is clear that when only these 400 features are taken into account, the performance of CatBoost is not affected much. Compared with the original CatBoost, the results of new CatBoost are even slightly improved and the results of the stratified 10-fold cross validation are better than that of the stratified 5-fold cross validation.

Table 3 Performance comparison of models in different stratified k-fold cross-validation experiments.

Models	ROC AUC	PR AUC	MSE	Pearson	
CatBoost (stratified 5-fold cross-validation)	0.9217	0.4651	0.1365	0.5335	
New CatBoost (stratified 5-fold cross-validation)	0.9208	0.4707	0.1360	0.5379	
New CatBoost (stratified 10-fold cross-validation)	0.9238	0.4860	0.1351	0.5466	

Discussion

This study employed the CatBoost model to predict the synergistic effects of anticancer drug combinations and comprehensively evaluated its predictive performance. Through stratified 5-fold cross-validation, it was found that CatBoost outperformed the other three models, including DNN, XGBoost, and logistic regression, across all evaluated metrics. CatBoost achieved a ROC AUC of 0.9217 and an MSE of 0.1365, demonstrating high predictive accuracy and stability.

Further analysis revealed that the correlation between the synergistic scores predicted by CatBoost and the actual values ranged from 0.52 to 0.83, indicating the model’s robust predictive capability for the synergistic effects of drug combinations across various cell lines. Additionally, by analyzing the distribution of PCC for both drugs and cell lines, no clear association was found between drug targets or cell line tissue types and PCC, suggesting that the model’s predictive performance is not influenced by these factors.

In terms of model interpretability, we utilized SHAP values to interpret the model and found that monotherapy information played a significant role in predicting synergistic effects. Moreover, several genes, including PTK2, CCND1, and ERBB2, were identified as being closely related to the occurrence and progression of cancer, with their expression levels significantly correlating with patient prognosis. In a comparative experiment through using only cell line features or only drug features, we found that drug features play a more prominent role in drug synergy prediction than cell line features.

To verify the impact of features with high SHAP values on model efficacy, we selected the top 400 features for a new prediction experiment. The results indicated that even with this reduced feature set, the performance remained largely unaffected and even slightly improved in certain metrics, suggesting that these features contain sufficient information to train a well-performing model.

In summary, the CatBoost model demonstrated excellent performance in predicting the synergistic effects of anticancer drug combinations and exhibited high feature interpretability. This study provides an effective predictive tool for the discovery of anticancer drug combinations, aiding in the acceleration of drug development processes and offering possibilities for future personalized medicine. Future research can further explore additional features and algorithms to enhance the accuracy and generalizability of predictions.

Conclusions

The good quality of our model for predicting drug synergy may be attributed to its several advantages. First of all, our model belongs to tree-based models. Compared to general DNN-based methods, these types of models are easier to build because they require less hyperparameter tuning or feature preprocessing.

Second, CatBoost uses the tactic of oblivious trees and ordered boosting to effectively avoid the problem of overfitting and improve the execution efficiency. Unlike the traditional decision trees, the oblivious trees have the advantages of small volume, high efficiency, and strong ability etc. Classical gradient boosting algorithms suffer from prediction bias and tend to overfit on small or noisy datasets. When computing gradient estimates for data instances, these algorithms use the same data instances on which the model is constructed, and therefore have no opportunity to encounter new data. Whereas CatBoost uses ordered boosting to combat noise points in the training set, thus avoiding bias in gradient estimation. Furthermore, CatBoost trains models and computes the residuals on different data subsets, thus it can prevent target leakage and avoid overfitting.

Third, this work used SHAP to make direct biological interpretation of the model output, which made it possible to do some more in-depth analysis of features. The results have indicated that the excellent performance of CatBoost is mainly due to the most important features, and at the same time, the speed of CatBoost operation could be greatly improved by reducing the feature dimension. We can assume that in the future, even when faced with very large data, CatBoost can downsize the feature space trough implementing SHAP and get a more stable model.

Finally, the performance and training time of the CatBoost model could be affected by the setting of hyperparameters as well as the size of the sample size. We believe that, with the increasing amount of data in the future, CatBoost will become more accurate and stable.

It is inevitable that our model has some limitations for drug synergy prediction. For example, using the synergy score to measure the therapeutic effect of drug combination may not be an ideal method as it is a score for a wide range of concentrations, but in practice, treatments with low concentrations perform better in the clinic. Additionally, we analyzed the forecasting effect of this model on different drugs or different cell lines, we found that the differences in prediction results were not significantly correlated with tissue-specific or target-specific mechanisms. These differences may be derived from the fact that some biological mechanisms are better modeled than others. The specific mechanism still needs to be further explored in the future.

Supplemental Information

Supplemental Information 1 The list of the top 100 features with SHAP values.

Additional Information and Declarations

Competing Interests

The authors declare that they have no competing interests.

Author Contributions

Changheng Li conceived and designed the experiments, performed the experiments, analyzed the data, performed the computation work, prepared figures and/or tables, authored or reviewed drafts of the article, and approved the final draft.

Nana Guan conceived and designed the experiments, authored or reviewed drafts of the article, and approved the final draft.

Hongyi Zhang performed the computation work, authored or reviewed drafts of the article, and approved the final draft.

Data Availability

The following information was supplied regarding data availability:

The data and code for this work are available at GitHub and Zenodo:

- https://github.com/AnnaGuan/CatBoost

- AnnaGuan. (2024). AnnaGuan/CatBoost: First release of CatBoost project (Version v1). Zenodo. https://doi.org/10.5281/zenodo.14189227.

The NCI-ALMANAC dataset is available at

NCI DTP Data: https://wiki.nci.nih.gov/display/NCIDTPdata/NCI-ALMANAC.

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
