# Peer review of "Anticancer drug synergy prediction based on CatBoost"

_PeerJ Computer Science, doi:10.7717/peerj-cs.2829_

## Round 0.1 · original submission · Major Revisions

The reviewers raised a number of issues that need to be address in a new version of the manuscript. The current version of the article has some flaws that should be solved. I invite the authors to prepare a new version of the article taking into account the recommendations provided by the reviewers.

Reviewer 1 ·

Basic reporting

The study presents a novel approach to predicting anticancer drug synergy using the CatBoost algorithm, leveraging features such as morgan fingerprints and gene expression profiles. The model demonstrated superior performance compared to other methods, and the use of SHAP provided insights into the biological relevance of the predictions.

Experimental design

Methods described lack details. How the new Catboost is constructed is not well defined. No details were provided about data preprocessing. The model selection methods were not adequately described.

Validity of the findings

The manuscript lacks an independent datasets for validation.

Cite this review as

Reviewer 2 ·

Basic reporting

Please see the comments in Additional Comment.

Experimental design

Please see the comments in Additional Comment.

Validity of the findings

Please see the comments in Additional Comment.

Additional comments

The authors proposed a machine learning approach based on the CatBoost algorithm to predict the synergy scores of anticancer drug combinations on cancer cell lines. Using data from the NCI-ALMANAC dataset, the study characterizes drugs with molecular fingerprints, target information, and monotherapy data, and characterizes cell lines with gene expression profiles. The CatBoost model was evaluated through stratified five-fold cross-validation and demonstrated superior performance compared to other models, such as deep neural networks (DNN), XGBoost, and logistic regression. Additionally, the Shapley Additive Explanations (SHAP) package was used to interpret the biological significance of the prediction results, identifying important features associated with known cancer mechanisms. However, I have several comments listed below:

Major comments:

1. What is the motivation for using the CatBoost method? What are the problems with existing methods? Why not use LightGBM?

2. DrugComb is a web-based database published in 2019 [1] and updated in March 2021 [2]. In addition to data on drug combination synergy, it also provides monotherapy sensitivity screening data for diseases such as cancers, malaria, and COVID-19. DrugComb offers a web server to analyze and visualize the synergy of drug combinations, predicting synergy scores for a given drug combination at a single dose level using a machine learning model, CatBoost. It also uses a drug-target network-based model to visualize the mechanisms of action of drug combinations. What is the difference between your method and DrugComb?

3. In this work, the authors optimized the threshold by implementing stratified five-fold cross-validation to achieve an optimal balance, setting the threshold value at 10. Is this threshold generalizable to other studies, or is it study-specific?

4. In Table 1, what is meant by the "new CatBoost"? What would the results be if 10-fold cross-validation were used? The improvements in the classification task of CatBoost are not significant compared to DNN. In the regression task, as mentioned, there are many samples with synergy scores close to 0, skewing the distribution. Therefore, PCC may not be an appropriate performance metric.

5. Line 247 mentions that "statistically, 88 out of 100 features belong to the drug." Does this imply that drug features play a more prominent role in drug synergy prediction than cell line features? What kind of statistical test do you use? There are 470 cell line features and 1594 drug features, thus it is natural to have more drug features in the top 100 features. It might be beneficial to compare the performance using only cell line features and only drug features to see any differences.

6. Why choose the top 400 features? Why not the top 100? Does this feature selection work for other data?

Minor comments:

1. Line 35, "add your introduction here". It seems the authors forgot to delete it.

2. Line 71, clarify the citation format: "Li et al. (Yousef et al. 2017)?"

3. Line 132 and Line 141 have different representations of 130182.

4. Line 194, the authors should specify the GPU and CPU. What is meant by a "normal work computer"?

Cite this review as

Reviewer 3 ·

Basic reporting

See below

Experimental design

See below

Validity of the findings

See below

Additional comments

Authors proposed a machine learning method based on CatBoost to predict the synergy scores of anticancer drug combinations on cancer cell lines, which utilized oblivious trees and Ordered Boosting technique to avoid overûtting and bias. The model was trained and tested using the screened data from NCI-ALMANAC dataset. The drugs were characterized with morgan ûngerprints, drug target information, monotherapy information, and the cell lines were described with gene expression proûles.
1. The abstract should be narrow down on the problem and highlight the need of the proposed work with experimental results.
2. Add the contents in the abstract of the paper and highlight the impact of the proposed work.
3. To explore Comparative results with existing approaches/methods relating to the proposed work.
4. The method/approach in the context of the proposed work should be written in detail.
5. In the introduction section, you should give the novelty and the contributions of your works.
6. The literature review is poor in this paper. You must review all significant similar works that have been done. Also, review some of the good recent works that have been done in this area and are more similar to your paper.
7. Result and discussion should be rewritten to summarize the findings/significance of the work.

Cite this review as

---

## Round 0.2 · Minor Revisions

Some improvements have been made by the authors but there are still some issues regarding the scientific literature review, that need to be addressed. I invite the authors to fix these problems and submit an improved version of the manuscript.

Reviewer 1 ·

Basic reporting

I reviewed the manuscript. I still have some concerns about the ethics of the manuscript. It reported some improved work from a previous publication by the same author. I think they should properly cite the reference: https://ijoer.com/assets/articles_menuscripts/file/IJOER-JAN-2024-2.pdf. And they should discuss the improvement in the current work. Unfortunately, they still failed to rewrite a comprehensive review of the current topic in the Introduction part. At least they missed some of the essential reference with the current topic, such as https://academic.oup.com/bib/article/23/1/bbab355/6363058, https://pubmed.ncbi.nlm.nih.gov/34060634/ (DrugComb), https://pmc.ncbi.nlm.nih.gov/articles/PMC10598574/, https://www.nature.com/articles/s41598-023-33271-3

Experimental design

They should deposit the original codes in public depositories so that other people can reproduce their results.

Validity of the findings

They should compare their results with previous publications.

Cite this review as

Reviewer 3 ·

Basic reporting

ok

Experimental design

ok

Validity of the findings

ok

Cite this review as

---

## Round 0.3 · accepted · Accept

The authors correctly addressed the remaining issues indicated by the reviewers, and therefore I can recommend the acceptance of the article.